# Healthcare Professionals' Perspectives on the Outgoing Geriatric Team: A Qualitative Explorative Study

**Sanne Have Beck** *[ID] **and Dorthe Susanne Nielsen** [ID]

Geriatric Research Unit, Department of Geriatric Medicine, Odense University Hospital,
University of Southern Denmark, 5000 Odense, Denmark
* Correspondence: sanne.beck@rsyd.dk

**Abstract:** The need for communication and collaboration increases when an older patient is discharged from the hospital, as the transition of care is complex for older patients living with multiple concurrent diseases. An intervention: The outgoing geriatric team was developed and initiated to address these patients' complex needs. The outgoing geriatric team aimed to collaborate with healthcare professionals at a skilled nursing facility. This study explored how the intervention was experienced by the healthcare professionals from both the outgoing geriatric team and the skilled nursing facility. The study employed a qualitative explorative design using semi-structured interviews. Fourteen healthcare professionals participated in the interviews. Data were analyzed using Braun and Clark's thematic analysis. Three themes emerged: (1) The need for personal contact and communication; (2) the need for competent care and sensitive observation; and (3) the need for clarification of responsibilities. The study emphasized the importance of meeting face-to-face during cross-sectoral collaboration when treating and caring for patients with complex care needs.

**Keywords:** cross-sectoral collaboration; outgoing geriatric team; skilled nursing facility

## 1. Introduction

The demographic trend toward an ageing population is predicted to increase significantly in most industrialized countries. At the beginning of 2021, people aged 80 years and older constituted approximately 5% of the Danish population. This figure is predicted to increase to 10% by 2050 [1]. A projected demographic profile has revealed that by 2050, people aged 70 years and older will double their days in the hospital and account for nearly 60% of all hospital days [2]. Despite these trends, there has been a reduction in the number of hospital beds [3] from 2007 to 2018 in an effort to increase efficiency within healthcare sectors across Europe.

Older medical patients are characterized as having several concurrent diseases, as well as declining physical and/or cognitive function. Due to the fact that these patients often live alone and have a decreased ability to take care of themselves, they frequently require care from primary or hospital services and heavy medical treatment [4].

When a patient with multiple and complex care needs is discharged from a hospital in Denmark, the municipality provides in-home or temporary care and treatment at skilled nursing facilities [4]. Transfer across care settings and care providers requires transitional care interventions and information for the patient and carers [5], as well as the involvement of the healthcare professionals [6]. It is vital for the involved healthcare professionals to communicate across the sectors before the patient is discharged. However, cross-sectoral electronic communication comes with challenges, such as insufficient or limited exchange of patient information, inconsistencies in communication, and the experience of not having access to all the needed information [7]. A study by Petersen et al. disclosed that nurses tended to focus on fulfilling the standard requirements for the cross-sectoral electronic communication system rather than focusing on the patients' needs and problems. The electronic communication form was also found to impede dialogue and communication [8].

The transitional care model (TCM) is a multicomponent, nurse-led intervention model [9] comprising nine components that target older adults moving across healthcare settings [10]. Components of the TCM have been used in various RCT studies, which also included older patients with multiple medical and care needs. These RCT studies showed that the TCM led to reductions in readmissions in some hospitals [9,11]. The TCM approach reportedly had a positive impact on the participants' quality of life [12] and led to reduced financial costs [12,13]. From a theoretical perspective, the theory of 'relational coordination' has similar components to those of the TCM. Relational coordination builds upon social networks [14], and in the positive cycle of relational coordination, shared knowledge, shared goals, and mutual respect are complemented by problem-solving communication that is frequent, timely, and accurate [14,15].

In Denmark, studies have been conducted on different interventions with outgoing teams from geriatric departments following up on patients discharged from either the emergency department or the geriatric departments to various locations [16–19]. These studies showed a reduction in readmission rates among nursing home residents [19] and also a reduction in readmission rates, compared to municipality-based interventions [16].

In 2018, the Department of Geriatric Medicine at a university hospital initiated an intervention, the out-going geriatric team (OG-team), which was developed based on literature and clinical experiences. The OG-team consisted of a multidisciplinary team of doctors and nurses working at the Department of Geriatric Medicine. The team arranged follow-up visits with patients discharged from the Department to temporary care at a skilled nursing facility (SNF) in an urban municipality. The multidisciplinary team visited the skilled nursing facility three times a week, and the face-to-face visits comprised a conference co-created by the healthcare professionals in the outgoing geriatric team (OG-Team), the healthcare professionals at the skilled nursing facility (SNF-team), and the patient. The purpose of these conferences was to make agreements on what issues should be addressed post-discharge. This study aimed to explore how the two teams experienced the cross-sectoral follow-up visit when the patient was discharged from the Department of Geriatric Medicine to a skilled nursing facility. Besides our study, the intervention was studied quantitatively with hospital readmission as an end-point [18].

## 2. Methods

### 2.1. Design

We conducted an exploratory qualitative study. The theoretical perspective and methodology was built on a phenomenological hermeneutic approach [20]. This approach made it possible to uncover the participants' perspectives and, during the analysis, allowed us to interpret their experiences. We used a semi-structured interview guide [21,22] to underpin the aim of the study.

### 2.2. Setting

Patients admitted to the Department of Geriatric Medicine are characterized by age of 65 years and older, acute medical illness, multimorbidity, polypharmacy, frailty, and functional and/or cognitive decline. These patients are most in need of comprehensive and holistic treatment and care in a setting where care needs are identified and addressed.

In Denmark, patients can be discharged from the hospital once they have been medically treated for their acute illnesses. However, older patients can be affected by acute illness and functional decline when they are discharged, and they often require more help and care than they did before hospitalization due to this decline. From the minute the patient is admitted to the hospital, the planning for discharge commences. During hospitalization, ongoing dialogues with the patient and relatives are conducted while the discharge is being planned. If the patient needs home care, the nurse is obligated to communicate with the healthcare professionals in the municipality to plan the discharge. When the patient needs substantial care and rehabilitation, the nurse will describe these needs in the cross-sectoral electronic communication report in collaboration with the patient and

his/her relatives. Therapists from the Department of Geriatric Medicine are responsible for the specific rehabilitation plan. It is up to the coordinator in the municipality to determine whether the patient is offered a stay at the skilled nursing facility for temporary care or is discharged to their own home. This determination is based on the municipality's prior knowledge of the patient and the cross-sectoral electronic communication report.

The healthcare professionals at the skilled nursing facilities comprise nurses and social and healthcare assistants. The purpose of a stay at a skilled nursing facility can be to either clarify the older person's home and care needs post-hospitalization or to rehabilitate the older person so that he or she can remain longer in his/her own home [23].

### 2.3. Participants

Participants were included in this study using purposive sampling [21], with the aim of inviting healthcare professionals with different types of education, employment length, and experiences with cross-sectoral collaboration to the outgoing geriatric team. In total, 17 healthcare professionals (all female) were invited. A gatekeeper at the skilled nursing facility made appointments with the included participants in the SNF-Team, and all agreed to participate. The average length of employment for the healthcare professionals at the skilled nursing facility was 4.9 years. Seven of the ten invited healthcare professionals from the OG-Team agreed to participate: six nurses and one doctor; three did not reply to the invitation. Their average length of employment in the Department of Geriatric Medicine was 7.4 years. For the purpose of anonymization, all participants were given an ID number see Table 1.

**Table 1.** Participants from the two teams, their profession, and length of employment.

| ID Number | Profession | Years of Current Employment |
|---|---|---|
| SNF-Team 1 | Social and healthcare assistant | 8 years |
| SNF-Team 2 | Nurse | 7 years |
| SNF-Team 3 | Nurse | 6 years |
| SNF-Team 4 | Nurse | 8 months |
| SNF-Team 5 | Social and healthcare assistant | 2 years |
| SNF-Team 6 | Nurse | 4 years |
| SNF-Team 7 | Social and healthcare assistant | 6.5 years |
| OG-Team 1 | Nurse | 5 years |
| OG-Team 2 | Nurse | 13 years |
| OG-Team 3 | Nurse | 13 years |
| OG-Team 4 | Nurse | 5 years |
| OG-Team 5 | Nurse | 1.5 years |
| OG-Team 6 | Doctor | 5.5 years |
| OG-Team 7 | Nurse | 9 years |

### 2.4. Data Collection

The interviews were conducted based on a semi-structured interview guide [21,22]. Interviews began with an opening question, where the healthcare professionals were asked about how they experienced the conference with the OG-Team and the cross-sectoral collaboration. Further questions were asked if the healthcare professional did not initially provide answers to questions such as: "What things were discussed at the conference"; "How are you involved in the collaboration"; and "What have you learned during the project period?". To get a deeper description of the phenomenon, prompts such as "What

do you think about . . . ?" or "What do you mean by . . . ?" and "Please, can you elaborate on this . . . ?" were asked.

The interviews were conducted between February 2019 and January 2020 by the author [SHB]. All interviews took place face-to-face in a setting chosen by the individual participant, either at work or at home. The interviews were audio-recorded and transcribed verbatim by the author [SHB].

### 2.5. Data Analysis

A thematic analysis was conducted using Braun and Clarke's theory, which focuses on individual explanations and socio-cultural contexts and structures [24]. Braun and Clarke describe six phases of analysis: (1) Familiarization with data: getting to know the data through transcribing and repeated reading and searching for meanings in the data. (2) Generation of initial codes: systematic work through data to generate an initial list of codes that appear interesting. Identifying codes that match with the data and also making sure to include descriptions about the context. (3) Searching for themes: in this phase, codes are sorted into potentially different themes, and an overarching theme will emerge from the coding. (4) Reviewing themes: in this two-step process, the themes are first reviewed in the context of the data that is described as extra. The themes are then read with the entire dataset. (5) Defining and naming themes: the story of each theme needs to capture the essence of what the theme is about. (6) The production of a report is the final step, which involves the task of telling a complicated story in a way that validates the analysis [24]. Data were analyzed by author [SHB] and author [DSN] using an inductive approach. We initially read the interviews at least twice to get familiar with the data. In the initial coding, we looked for patterns across all data, marking them with different numbers. The patterns were then grouped into themes. We used an iterative process to review the themes, going back and forth between the themes, codes, quotes, and interviews in order to ensure that the data underpinned the themes, see Table 2.

**Table 2.** Example of analysis from quotes to theme.

| Quotes | Initial Codes | Searching for Themes | Defining and Naming the Theme |
|---|---|---|---|
| "They listen and take what I say seriously" "They try to learn our names" "Focus on the patient" "I can ask about other things as well" | Meeting each other Collaboration Communication | Professional relationship Getting to know each other | The need for personal contact and communication |

### 2.6. Ethical Considerations

In accordance with the Helsinki Declaration [25], informed consent was obtained for participation. The study was approved by the Ethical Committee in the Region of Southern Denmark (reference number 20182000-160), and participants were informed in writing and verbally about the aim and method of the study before written consent was obtained. Participants were informed of their right to withdraw from the study at any time without prejudice toward their work. Confidentiality and anonymity were protected, and data were stored according to the guidelines of the Danish Data Protection Agency (reference number 20/28195).

### 2.7. Findings

The analysis revealed three themes. The first theme, "The need for personal contact and communication", revealed that the participants found it important to get to know each other to create a room for collaboration where all knowledge was equally acknowledged. The second theme, "The need for competent caring and sensitive observation", illustrated that shared knowledge about older patients had an impact on the health professionals'

competencies. The last theme, "The need for clarification of responsibilities", described the challenges experienced with cross-sectoral collaboration.

### 2.7.1. Theme 1: The Need for Personal Contact and Communication

All participants expressed that the outgoing geriatric team intervention had a positive impact on the cross-sectoral collaboration surrounding the older patient when he/she was discharged to the skilled nursing facility.

Most participants found the physical face-to-face meeting at the skilled nursing facility was the best way to collaborate cross-sectorally and consult with the patient. Most participants expressed that knowing each other across professions and sectors and building up a relationship proved vital to the quality of their collaboration. Most participants underlined that knowing each other by name and profession led to more straightforward, trustful, and open communication, where questions could be asked and doubts could be shared without the feeling of being "looked down on" or "stigmatized". Furthermore, most participants acknowledged that all competencies and information were equally important, respected, and acknowledged.

> *"I don't experience any hierarchy. We are 'in the same boat' and each of us contributes with knowledge about the patient"* (OG-Team 4)

The cross-sectoral collaboration gave some participants the feeling of being part of a team. They experienced that all healthcare professionals were dedicated, and that they focused on the patient's needs; the participants expressed that it was for the best in most cases for the patient to stay at the skilled nursing facility or choose to start treatment for an infection in order to reach a common understanding of what additional observation and care should be initiated.

The collaboration relied on all participants being well-prepared before the conference to be able to reflect on the patient's status. The shared reflections provided all participants with responsibility for the collaboration. Furthermore, the collaboration ensured a consensus about the medical plan for the patient.

> *"I experience that the patient isn't lost in the transition between the hospital and the skilled nursing facility"* (SNF-Team 3)

Participants from both teams expressed that working together cross-sectorally demanded preparation before the conference. The degree of preparation and knowledge of the patient's personal and medical history affected the dialogue and feedback. Lack of time for preparation was described as frustrating by most participants and had a negative impact on the outcome of the consultation.

> *"Being prepared is important but not always possible"* (SNF-Team 4)

> *"If I don't have the time to read and prepare, there is not much dialogue, but more one-way communication"* (OG-Team 6)

### 2.7.2. Theme 2: The Need for Competent Care and Sensitive Observation

The vulnerability and complexity of patients were underlined as a challenge to all participants in the SNF-team, especially in acute situations where most of the participants from the SNF-team felt insecure and afraid of not being competent enough to help the patient. The collaboration between the two teams enabled the SNF-team to receive qualified advice about and feedback on the patient's condition and made it possible for the patient to remain secure in their environment at the skilled nursing facility.

> *" . . . you have to take things in the stride, so the patient doesn't have to go to the general practitioner or acute medical unit and back and forth"* (SNF-Team 2)

The healthcare professionals at the skilled nursing facility sought to identify home and care needs for the patient after a hospital stay. Most participants from the SNF-team experienced that the OG-team not only focused on the medical treatment of the patient

but also shared a more holistic approach. This holistic approach toward the patient was cited as a common aim that helped motivate all participants to stay focused on the patient's special wishes for the future, the need for homecare, and future hospitalization, as well as the patients' wishes for future treatment and care.

*"They ask about the patient, they don't just focus on the patient's disease"* (SNF-Team 3)

The cross-sectoral collaboration gave the SNF-team a plan to act upon and knowledge about how to handle different situations together with the patient.

All participants underlined the importance of the synergy of knowledge, skills, and competencies growing between the two teams. This improved the quality of the treatment and care provided for the patient, with some participants expressing that they prevented readmitting the patient to the hospital and were able to continue focusing on the individual patient's wishes and needs.

*"Transition from the hospital is hard for the vulnerable older patient. If I get concerned while observing him/her, I talk to the outgoing team (and ask) what do you think? Should I be concerned? . . . Feedback gives me new knowledge and helps me understand why the patient reacts in certain ways . . . I think that we have avoided having to readmit the patient to the hospital and moving (the patient) back and forth due to our insecurity"* (SNF-Team 2)

The SNF-team emphasized that collaboration made it possible to get feedback on the care and treatment that were provided, which made them feel that they had deepened their knowledge on geriatrics and become more competent at observing and attending to the care needs of this patient group.

*" . . . if the healthcare professionals at the skilled nursing facility say she (patient) doesn't drink that much, I hardly have to say a word before they say, 'Should we start a fluid schedule?'"* (OG-Team 7)

Both teams expressed that through their collaboration, they were able to establish a foundation where they could help each other, on-site, to make more competent observations and to deliver improved care and treatment.

2.7.3. Theme 3: The Need for Clarification of Responsibilities

In the collaboration between the OG-team and the SNF-team, two challenges were identified.

During the interviews, all participants disclosed that they were sometimes uncertain about who was responsible for the patient on days when the OG-team was not at the SNF. Some healthcare professionals in the OG-team felt that they were not accessible enough outside of the actual conferences, while others deemed it unacceptable for the SNF-team to contact them on days when no conferences were held. Some participants from the SNF-team felt free to contact the OG-team during the daytime. Others in the SNF-team expressed that they lacked guidelines on how to handle acute situations when the OG-team was not around, which could cause some of the healthcare professionals to worry and cause insecurity.

*"They often call us if they have questions, which is good for the collaboration"* (OG-Team 3)

Participants also experienced a lack of clear distribution of roles and responsibilities between the OG-team and the patient's general practitioner (GP). This caused increased confusion for some of the participants in the SNF-team because in an acute situation, they were sometimes in doubt about who to contact. Some participants from the SNF-team experienced the GP refusing to consult the patient because the patient was already being seen by the OG-team.

*"What issues are handled by the outgoing geriatric team and which ones by the GP? Who has the responsibility? Sometimes I feel no one is responsible"* (SNF-Team 4)

*"Where is the boundary? When are we to take care of it and when is it the GP? I experience that sometimes the healthcare professionals at the skilled nursing facility wait to contact the GP or doctor from the emergency department if they think the outgoing geriatric team can handle the issue"* (OG-Team 6)

## 3. Discussion

First of all, our study revealed that the face-to-face encounter and collaboration facilitated the teamwork and provided the healthcare professionals with a trustful opportunity to give and receive feedback when consulting. Similar results have been described in a study involving healthcare professionals, nurses from community healthcare, and general practitioners in Turkey [6]. The study found that effective communication across healthcare settings could reduce readmission rates [6], as they found that the letter of discharge had often failed to include follow-up information on treatment started in the hospital, which resulted in an interrupted chain of care for the older patients with complex care needs [6].

Our study underlines the importance of constructing strong collaboration and teamwork cross-sectorally when the patient is dependent on care and assistance from many health professionals. The Transitional Care Model's nine components are: screening; staffing; maintaining relationships; engaging patients and caregivers; assessing/managing risks and symptoms; educating/promoting self-management; collaborating; promoting continuity; and fostering coordination [10]. The collaboration in our study had similarities to the 'Fostering Coordination' in the TCM. In our study, the conference was the place where the healthcare professionals achieved shared understandings and goals for the treatment of the older patient, as they had a shared responsibility. We also found that in the face-to-face collaboration, together with patients and relatives, the patient's symptoms could be discussed and put into context to give the healthcare professionals the knowledge and competencies to act. Maintaining relationships, continuity of the healthcare professionals in the two teams, and coordination of tasks and feedback was supported by the heads of departments, as the intervention was highly prioritized by all.

The intervention in our study differed from the TCM in some ways. First of all, it was not nurse-led but involved an interdisciplinary team-led intervention. Secondly, for some patients, one visit from the OG-team was sufficient and had the impact of reducing readmission rates [18]. Thirdly, no screening [10] was performed, as all patients discharged from the Geriatric Department had multiple risk factors. Just as our study showed interventions that had similarities to components in the TCM, the study by Burke et al., (2014) revealed that the specific components (assessing/managing risk and symptoms; fostering coordination; and education/promoting self-management) had a significant effect on reducing readmission rates [26].

The intervention of the current study, the outgoing geriatric team, was further studied quantitatively by Thomsen et al., (2021) in a before-and-after cohort study. Their results showed a 28% reduction in hospital readmissions within 30 days after the patient was discharged to the skilled nursing facility [18]. These findings support our qualitative findings that show the healthcare professionals' experiences of fewer patients being readmitted to the hospital after the implementation of the cross-sectoral physical consultation.

The cross-sectoral collaboration we found in our study can also be deduced from the theory of relational coordination [14]. This theory was developed in airline and healthcare research, where multiple individuals work together in complex systems [27]. Relational coordination builds upon social networks and social capital [14]. In the positive cycle of relational coordination, shared knowledge, shared goals, and mutual respect are complemented by problem-solving communication that is frequent, timely, and accurate [14,15]. In relational coordination, each person acts interdependently rather than independently to achieve the best outcome through the transfer of information and resources [14]. In our study, we found that the cross-sectoral collaboration surrounding the older patient was challenged and depended on the individual healthcare professional's competencies in observing and taking action, as well as on their ability to communicate and collaborate. In

the cross-sectoral collaboration between the SNF- and OG-teams, shared knowledge was emphasized to be significant in the care and treatment of older patients. The teams created cohesion where they, along with the patient, had shared goals and mutual respect.

We identified some challenges in the study. There was no clear definition of who was responsible for the patient when the OG-team was away from the nursing facility. The participants from all teams also experienced that there was no clear agreement on what the OG-team could be contacted about on the days they were not at the skilled nursing facility or how and when the SNF-team should contact the patient's GP. A review by Janssens et al., (2020) explored competencies among doctors' collaboration skills in a primary–secondary interface [28]. Among these competencies, 'collaborative attitude and respect', 'roles and responsibilities', 'mutual knowledge and understanding', 'communication', and 'leadership' were identified and emphasized to provide good collaboration. On the other hand, unclear roles and responsibilities were described as barriers to good collaboration [28], which accentuates the findings in our study.

We found that the participants in our study experienced an increased level of competence in taking responsibility and caring for older patients. In a Norwegian community, a self-assessment survey examining healthcare professionals' competencies in caring for older people was conducted, and it revealed that the nurses' levels of care competencies varied across age, but also across workplaces where nursing staff scored higher at nursing homes than staff in home care [29]. Thus, in a skilled nursing facility, it would be relevant to increase the focus on the healthcare professionals' clinical caring competencies, given the complexity of older patients' treatment and care needs. Our findings emphasize that the cross-sectoral collaboration and meeting face-to-face led to increased knowledge among both teams and an experience of improved competencies, patient treatment, and care.

*Strengths and Limitations*

To our knowledge, this is the first qualitative study in Europe to explore the perspectives of healthcare professionals in a cross-sectoral collaboration with an multidiciplinary outgoing geriatric team and healthcare professionals at a skilled nursing facility. The study was conducted as an explorative interview study, allowing the healthcare professionals to elaborate on their experiences within the intervention. We chose to conduct individual interviews, where the participants were able to elaborate on their experiences, and we in turn could explore their thoughts and spoken words. Our study could have been enriched, however, by conducting onsite field observations focusing on the healthcare professionals' relations, actions, and interactions.

A potential limitation to the study is that the healthcare professionals in the OG-team and the first author were colleagues at the time, which could have rendered them less critical during the interview. This was accommodated by ensuring the participants' anonymity before beginning the interview and exploring both the positive and negative statements. It was not examined further whether this was the reason that three participants did not reply to the invitation to participate in the interview. Another limitation is that only one doctor participated in the study, and this could have downplayed the insights of that profession.

Lastly, our study was performed on a small scale within only one geriatric department and one SNF in one municipality over a year. Changes such as replacement of healthcare professionals and education level might have influenced the clinical practice and evolvement of the environment of the intervention, but these factors were not taken into account in this study.

## 4. Conclusions

This study emphasized the importance of meeting face-to-face when caring for and treating patients with complex care needs. The collaboration gave the healthcare professionals an experience of having increased competencies when working interdependently. The study also contributed important insights into how roles and responsibilities in cross-

sectoral collaboration need to be clearly defined and agreed upon. Collaborating across sectors is beneficial to all involved health professionals. The two teams in our study became a mutual team working with the patient and thereby creating the foundation for a coherent patient course. Future studies might explore whether these insights could be transferable to other care settings.

*Implications for Practice*

Cross-sectoral collaboration becomes important when treating and caring for the older patient, who may still have multiple and complex care needs. The need for face-to-face communication for the healthcare professionals in cross-sectoral collaboration as a supplement to written communication proved to be beneficial for the healthcare professionals both in the OG- and SNF-teams and increased the experience of having greater competencies.

**Author Contributions:** Conceptualization, S.H.B. and D.S.N.; methodology, S.H.B. and D.S.N.; data collection, S.H.B.; analysis, S.H.B. and D.S.N.; writing—original draft preparation, S.H.B.; review and editing, S.H.B. and D.S.N. All authors have read and agreed to the published version of the manuscript.

**Funding:** This research received no external funding.

**Institutional Review Board Statement:** Confidentiality and anonymity were protected, and data were stored according to the guidelines of the Danish Data Protection Agency (reference number 20/28195). The study was conducted in accordance with the Declaration of Helsinki, and approved by the Ethical Committee in the Region of Southern Denmark (reference number 20182000-160, date of approval 18 December 2018).

**Informed Consent Statement:** Informed consent was obtained from all subjects involved in the study.

**Data Availability Statement:** The dataset generated during the current study are not publicly available due to ethical conciderations related due to confindentiality. Data are avaliable for both authors.

**Acknowledgments:** The authors would like to thank all participants for taking the time to share their experiences and perspectives in this study.

**Conflicts of Interest:** The authors declare no conflict of interest.

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
