# Peer review of "Healthcare Professionals’ Perspectives on the Outgoing Geriatric Team: A Qualitative Explorative Study"

_2673-9259, doi:10.3390/jal2040026_

Round 1
Reviewer 1 Report
Compact and well-summarized. I have several suggestions to make your argument clearer.
<Abstract>
An abstract usually does not use headings (e.g., method, results, conclusion)
<Writing>
#1. When you use abbreviations for the first time, please spell them out. e.g., SNF, OG, GP
#2. 3rd paragraph on p. 2 (The one starting with "In Denmark") is too long.
#3. Aim - Two teams. How are they different? One is "interdisciplinary," right?
"Old" - 65 and older?? Please define
#4. Findings - G4, L3 -- I assume these are your interviewees. But, unless you provide some background information for each interviewee (age, gender, ESPECIALLY years of experience ad speciality area), they are meaningless.
#5. Discussion - Cite (6) right after Turkey.
#6. Strength and Limitation - "To our knowledge, this is the first ...." Is it true? Your references are mostly studies conducted in Europe. Am I right on this? If I am right, you should say, "this is the first qualitative .... in Europe" If your literature review covered the whole world, please make it clear in the beginning of the article.
Author Response
Thank you for all the relevant comments, we have tried our best to follow and elaborate on the issues raised by all three reviewers. As recommended by one reviewer we have now had a native English-speaking proofreader to revise the manuscript. We very much hope you will find the paper ready for publication.
Once again thank you for your interest in our paper.
Please see the attachment

Reviewer 2 Report
Please see attached file

Author Response

(The authors gave the same response as above.)

Reviewer 3 Report
Dear authors,
Thank you for the opportunity of reading your research results. The study brings an original contributions to the field of Gerontology.
The topic of the paper is very well and originally approached, but the structure of the paper lacks clarity. The theoretical background section is not presented. I identified the theory of Relational Coordination in the discussion section of the paper, the theoretical approach should be clearly presented right after the introduction. Why does this study bring contribution to this theory? and what is the research gap that this study is covering? How can you link the Cross-sectoral electronic communication and the transitional care model to this theory? These should be a few questions authors should respond to in the theoretical background section.
The aim should be presented in the introduction and should be more elaborated including also expected results.
The methodology includes the presentation of two methods phenomenological hermeneutic approach and a semi-structured interview guide. Could authors offer more details about how these two methods were applied in the previous similar research? and why are these two methods suitable for this study?
The results and discussion are clearly presented. Conclusion lacks theoretical implication, research limitations and future direction of research.
Editing for the abstract and references is mandatory.
Author Response

(The authors gave the same response as above.)
